# The Prevalence of Falls Among Older Adults Living in Long-Term Care Facilities in the City of Cape Town

**DOI:** 10.3390/ijerph22030432

**Published:** 2025-03-14

**Authors:** Nabilah Ebrahim, Jaron Ras, Rucia November, Lloyd Leach

**Affiliations:** Department of Sport, Recreation and Exercise Science, Faculty of Community and Health Sciences, University of the Western Cape, Robert Sobukwe Rd, Bellville, Cape Town 7535, South Africa; rasj@cput.ac.za (J.R.); novemberru@cput.ac.za (R.N.); lleach@uwc.ac.za (L.L.)

**Keywords:** ageing, falling, geriatrics, nursing homes, elderly

## Abstract

Falls are a prevalent health concern with a multi-factorial origin and causing numerous complications for older adults, especially those in low- and middle-income countries, such as South Africa. This study aimed to determine the prevalence of falls among older adults living in different types of long-term care facilities in the City of Cape Town. A total of 258 males and females aged between 60 and 95 years were recruited. Data collection occurred from September 2021 to January 2022. Participants were categorized into low, moderate, or high fall risk using the fall-risk assessment tool. Descriptive statistics summarized participant characteristics and fall prevalence. The chi-squared test determined significant associations between facility type, marital status, and educational level, and their relationship to falling. Spearman’s rank correlation coefficient assessed associations between fall occurrence and various risk factors. Findings indicated that the prevalence of falls was 32.6%. Falls were significantly associated with behaviors such as agitation or confusion, and other unmentioned risk factors. Participants living in facilities run by non-profit organizations, and who had a lower education level or were single or widowed, had a higher fall prevalence. The use of anti-depressant and anti-diabetic medications was significantly linked to increased fall risk. These findings can inform future research and the development of strategies to prevent falls among older adults, particularly in LTC facilities.

## 1. Introduction

The likelihood of falling increases significantly with age, particularly among those residing in long-term care (LTC) facilities, where the risk is even higher. Residents in LTC facilities are generally older, tend to have co-morbidities, need more support in navigating ADLs, often take more medication, and are considered physically and cognitively weaker compared to those living in the community [1]. The elderly living in LTC facilities are also highly dependent on caregivers, who are already overburdened with additional responsibilities [2]. Despite these high demands on caregivers, additional professional services should be provided for the elderly in these LTC facilities [2]. This elevated risk persists despite some evidence suggesting improvements in the overall health of older adults [3].

Globally, the population aged 65 years and older was approximately 703 million in 2019, with this number projected to double by 2050 [4]. In South Africa, the elderly population is also growing rapidly, particularly in provinces like the Western Cape, where the percentage of older adults increased from 8.9% in 2011 to 10.7% in 2022 [5]. This demographic shift is particularly pronounced in developing countries, where aging is occurring at an accelerated rate, compounded by socioeconomic challenges such as poverty, and exacerbated by the HIV/AIDS and COVID-19 pandemics [4,6,7].

In sub-Saharan Africa, and especially in low- and middle-income countries (LMICs), the older population is expanding at a faster pace than in developed nations, often without the necessary infrastructure and resources to provide adequate care [6]. This is concerning, as the highest proportions of older adults in South Africa are concentrated in regions like the Eastern Cape (11.5%), Western Cape (10.7%), and Northern Cape (10.1%) [5]. Given these trends, understanding and mitigating the risk factors for falls in this vulnerable population is crucial.

Falls among older adults are typically not caused by a single factor, but rather by a combination of intrinsic (e.g., visual problems, osteoporosis, balance difficulties), extrinsic (e.g., environmental hazards), and behavioral factors [8]. These multiple risk factors collectively increase the likelihood of falling, particularly in LTC facilities where older adults often experience reduced mobility and frailty. Addressing these risk factors is essential for fall prevention and improving the quality of life for older adults [9,10].

The consequences of falls extend beyond physical injuries; they also include significant psychological impacts such as loss of confidence, fear of falling, and limitations in activities of daily living (ADLs). These consequences can lead to a decline in functional capacity and overall quality of life [11]. Given the severity of these outcomes, it is imperative to develop targeted interventions to reduce fall risks in LTC facilities.

Despite the growing elderly population and the associated risks, there has been a lack of research specifically investigating the prevalence of falls among older adults in LTC facilities in the City of Cape Town. Older adults living in long-term care facilities in the City of Cape Town have a high prevalence of falls, with factors such as age, mobility, medication use, and environmental conditions contributing to the frequency and severity of these incidents. This gap in the literature underscores the need for studies that can inform local healthcare practices and policies. Therefore, this study aimed to determine the prevalence of falls among older adults living in various LTC facilities in the City of Cape Town, and it provides valuable insights for future fall-prevention strategies.

## 2. Materials and Methods

This study used a cross-sectional design where 258 male and female older adults were recruited to participate in the study. Due to the COVID-19 pandemic, limitations on sample size were imposed by the care facilities. Potential participants were identified by the nursing managers and, thereafter, an information session about the research was held with the potential participants. The 15 facilities that consented to participate in the study were regions in the Atlantic Seaboard, Southern Suburbs, and the Northern Suburbs of Cape Town. Data collection was conducted on a once-off basis from September 2021 to January 2022. Males and females, aged 60 years and older, who had been living in LTC facilities in the CoCT for a minimum of one year, and who were independently ambulatory, without the support of a walking aid, were included. Older adults who were not living in LTC facilities and those with physical disabilities who were dependent on a walking aid for ambulation, were frail, wheelchair-bound, or bed-ridden, were excluded. Ethical clearance to conduct the study was provided by the Biomedical Research Ethics Committee of the University of the Western Cape (Ethics number: BM21/6/18).

### 2.1. Research Procedure

The participants who consented to participate in the study then indicated the times and days on which they would complete the battery of tests. Each LTC facility had a designated rehabilitation or therapy room. This was used to conduct the tests. Testing of the participants was done on an individual basis, with the researcher and research assistants adhering strictly to the WHO COVID-19 protocols. A researcher-generated questionnaire was used to obtain the participants’ sociodemographic information and physical measurements, such as age, gender, educational level, marital status, and medical history. The participants were initially familiarized with the physical assessments, before proceeding with the actual testing. Participants were asked to complete a consent form before any data were recorded. Participation ranged from filling in a questionnaire to participating in risk assessments of a physical nature to gather relevant research information. The researcher and research assistant conducted the physical tests and gathered information via the questionnaires. A staff member at the LTC facility was always present to assist and make sure that when the tests were being conducted, participants felt safe and secure.

### 2.2. Fall-Risk Assessment Tool

The fall-risk assessment tool FRAT is a previously validated 4-item assessment tool used in residential care settings and is composed of two sections. Part one is a screening tool used to obtain a risk score [12]. Questions are asked regarding recent falls, medications used, and the participant’s psychological and cognitive status. The total score for the fall-risk status indicates a low (5–11), moderate (12–15), or high fall-risk (16–20) for the participant. The second part of the FRAT is a risk-factor checklist to identify possible risk factors that may contribute to falling. These are identified as major risk factors for falls in hospitals and residential care facilities, and include poor vision, concerning behaviors (e.g., agitation, confusion, disorientation, and difficulty following instructions/non-compliance), unstable mobility, unsafe transfers, incontinence, unsafe footwear, a challenging environment, poor nutrition, and impaired ADLs and other risk factors not listed in the study. According to Stapelton et.al (2009), the FRAT is a reliable and moderately accurate fall-risk screening tool for use in subacute and residential aged care facilities. They further concluded that the assessment tool had high inter-rater reliability [ICC (2,1) = 0.79] [13].

### 2.3. Data Analysis

Data were analysed using IBM SPSS Statistics (version 28). The data were checked for normality using the Shapiro–Wilks test. Descriptive statistical analysis (means, standard deviations, and frequencies) was used to describe variables such as age and BMI. Spearman’s correlation was used to determine the associations between falls and fall-risk factors. The Pearson’s chi-squared test was used to determine the associations between categorical variables (facility type, gender, age group, marital status, and education level).

## 3. Results

A total of 357 participants initially consented to participate in the study; however, due to constraints imposed by the COVID-19 pandemic, only 258 participants were recruited, resulting in a response rate of 72.3%. Of these participants, 28.7% (*n* = 74) were male, and the majority, 71.3% (*n* = 184), were female (Table 1). The mean (X̅ ± SD) age of the participants was 78.2 ± 7.4 years. The age distribution showed that 13.2% (*n* = 34) of participants were in the 60–69 years age group, 39.9% (*n* = 103) were in the 70–79 years age group, 43.0% (*n* = 111) were in the 80–89 years age group, and 3.9% (*n* = 10) were aged 90 years or older.

The results indicated that most participants (52.3%; *n* = 135) were either overweight or obese. In terms of education, 50.4% of participants had completed matric (grade 12), with 30.6% (*n* = 79) from mainstream schools and 19.8% (*n* = 51) from technical schools. Additionally, 20.5% of participants held a graduate qualification from a tertiary institution. Almost half of the participants (43.4%; *n* = 112) were widowed.

The study found that 32.6% of participants had experienced a fall in the past three months, while 67.4% (*n* = 174) had not sustained a fall during this period. The primary mechanisms of falls were slipping/tripping (15.5%; *n* = 40), followed by loss of balance (10.5%; *n* = 27) and dizziness (5.0%; *n* = 13).

On examining the three-month fall prevalence, it was found that females had a higher prevalence of falls (16%; *n* = 31) compared with males (12%; *n* = 9), with slipping/tripping being the most common cause. Conversely, dizziness was more prevalent among males (10%; *n* = 8). Loss of balance showed a similar prevalence among males (10%; *n* = 8) and females (10%; *n* = 19). Of the 40 participants who experienced slipping/tripping, 30.0% (*n* = 12) were in the 70–79 years age group and 45.0% (*n* = 18) were in the 80–89 years age group. Similarly, of the 27 participants who experienced loss of balance, 40.7% (*n* = 11) were in the 70–79 years age group and 48.1% (*n* = 13) were in the 80–89 years age group.

The prevalence of fall-risk factors among older adults showed that poor vision was the most common (49.6%; *n* = 128). Other notable risk factors included concerning behaviors (20.5%; *n* = 53), poor mobility (15.5%; *n* = 40), unstable transfers (11.2%; *n* = 28), incontinence (9.7%; *n* = 24), and various unspecified factors (2.7%; *n* = 7).

The fall-risk factors by gender (Figure 1) revealed that poor vision was the most prevalent among both males (44.6%; *n* = 33) and females (51.6%; *n* = 95). Concerning behaviors were reported similarly by males (20.3%; *n* = 15) and females (20.7%; *n* = 38). The prevalence of unstable mobility was comparable between males (15.2%; *n* = 12) and females (16.2%; *n* = 28). Unsafe transfers were slightly more common in males (12.2%; *n* = 8) compared to females (10.9%; *n* = 20). Incontinence as a risk factor was slightly higher in females (10.3%; *n* = 19) than in males (8.1%; *n* = 5). Unsafe footwear presented with a similar prevalence in males (5.4%; *n* = 4) and females (6.0%; *n* = 11). Environmental challenges were reported by 2.7% (*n* = 2) of males compared to 4.3% (*n* = 8) of females. Poor nutrition was more prevalent in females (4.89%; *n* = 9) than in males (2.70%; *n* = 1). Impaired ADLs were slightly more common in males (4.1%; *n* = 2) than in females (2.71%; *n* = 5).

Regarding age groups (Figure 2), participants aged 60–69 years had a high prevalence of risks related to challenging environments (8%; *n* = 3), poor mobility (26%; *n* = 9), and unsafe transfers (17%; *n* = 6). In the 70–79 years age group, unsafe footwear (0.9%; *n* = 1), incontinence (10%; *n* = 11), and poor vision (52%; *n* = 54) were the most common risk factors. The 80–89 years age group showed a high prevalence of poor nutrition (6.3%; *n* = 7), unspecified risk factors (3.6%; *n* = 4), and impaired ADLs (3.6%; *n* = 4). Among those aged 90 years and older, impaired ADLs (20%; *n* = 2), unsafe footwear (10%; *n* = 1), and unstable mobility (20%; *n* = 2) were the most prevalent.

The relationship between the sociodemographic characteristics of the participants and the occurrence of falls (Table 2) revealed that facility type (X^2^ = 7.403; *p* = 0.007), level of education (X^2^ = 14.05; *p* = 0.029), and marital status (X^2^ = 16.49; *p* = 0.001) were significantly associated with falls. The analysis also indicated a significant association between falls and certain observed behaviors such as concerning behaviors (X^2^ = 6.486; *p* = 0.011) and other issues (X^2^ = 4.951; *p* = 0.026).

Regarding the number of medications used, 62.0% of participants were taking more than two medications, with the majority being female (71.3%). The highest medication usage was observed in the 70–79 years (42.5%) and 80–89 years (39.4%) age groups. A total of 461 medications (Figure 3) were used by the participants. The smallest percentages of medications used were anti-Parkinson’s medications (2.3%; *n* = 6), anti-epileptic medications (2.7%; *n* = 7), and anti-psychotics (3.5%; *n* = 9). Sedatives (6.2%; *n* = 16), anti-diuretics (11.6%; *n* = 30), anti-diabetic medications (14.0%; *n* = 36), and anti-depressants (17.4%; *n* = 45) were more commonly used. The majority of participants were medicated with anti-hypertensives (58.9%; *n* = 152) and other medications (60.5%; *n* = 156). Participants who used anti-depressants [X^2^ (1) = 4.941; *p* = 0.026, OR = 2.083 (95% CI: 0.611, 1.763)] and anti-diabetic medications [X^2^ (1) = 4.097; *p* = 0.043, OR = 2.070 (95% CI: 1.013, 4.228)] were approximately four times more likely to experience a fall.

## 4. Discussion

The study aimed to determine the prevalence of falls among older adults living in LTC facilities in the City of Cape Town. The findings revealed a fall prevalence of 32.6% among participants, aligning with similar studies in Brazil (38.9%), Saudi Arabia (34%), China (31.7%), and Malaysia (32.8%) [11,14,15,16]. However, these results differ from studies conducted in the United Arab Emirates (UAE) and Egypt, which reported higher prevalence rates of 50.8% and 60.3%, respectively [10,17]. Another study indicated that fall prevalence was higher among women than men, which is consistent with this study’s findings [18]. On the other hand, a systematic review showed a lower global prevalence of falls (26.5%) [19]. These variations in fall prevalence could be influenced by several factors, including differences in study design, cultural norms, and healthcare conditions across countries.

Falls related to slipping/tripping showed a much higher prevalence of 68.5% [20]. Factors such as poor mobility, incorrect footwear, decreased strength, and balance issues may explain the high incidence of falls due to slipping/tripping [1,21]. The majority of falls occurred among participants aged 70–79 years (39.9%) and 80–89 years (43.0%), consistent with other studies where the highest fall rates were also observed in the 80–89 age group, particularly in LTC facilities [1,22]. The lower fall rates among participants aged 60–69 years may be attributed to their relatively better physical strength [22]. Participants aged 90 years and older often experienced lower fall rates, possibly due to being bedridden or less mobile, which reduced their fall risk [22]. This contrasts with findings from other studies, which suggest that age is not always a significant determinant of falls [22,23].

A significant association was found between concerning behaviors and falls in this study. Nearly half (47.2%) of the participants exhibited problem behaviors, which significantly increased fall risk. These behaviors may be influenced by the participants’ perceptions of independence, their fall history, their understanding of fall risks, and their willingness to accept support from staff and healthcare professionals [24]. In LTC facilities, safety regulations are often enforced by staff, which can lead to risk-taking behaviors among residents as they adapt to institutional rules, ultimately increasing fall risk [25,26,27]. The study also found that other risk factors not listed as major factors, were significantly associated with falls. Multiple studies have identified over 400 risk factors for falls, with no universally accepted classification system to simplify their understanding [28,29,30]. According to Williams et al., various factors, including older age, female gender, physical frailty, muscle weakness, poor gait and balance, impaired cognition, and depressive symptoms, contribute to an increased risk of falls. This risk further escalates with comorbidities such as cardiovascular disease, arthritis, and diabetes [31]. The study revealed that participants from NPO LTC facilities were more likely to experience falls than those from private facilities. This may be due to understaffing, limited safety resources, cost-cutting measures, or inadequate infrastructure in NPO facilities [32,33,34]. In contrast, private LTC facilities often employ multidisciplinary teams, including occupational therapists, physiotherapists, and biokineticists, who specialize in working with older adults and implementing fall-prevention procedures. These professionals also provide individualized care and continuously educate staff on safety measures [35,36]. Unfortunately, public healthcare facilities for older adults are often underfunded and must contend with a high burden of disease, limited support staff, and overcrowding compared to private facilities [36].

Education level was also significantly associated with fall risk in this study. Participants with lower education levels were more likely to experience falls, consistent with other research showing that individuals with lower education levels, particularly women, have higher fall rates [37]. It is suggested that lower education levels may lead to difficulties in understanding fall-prevention information, indicating the need for targeted fall-prevention programs among this population [38]. Additionally, lower education levels may affect cognitive abilities, making tasks like visual search more challenging and increasing fall risk [39].

Regarding marital status, the study found that married participants had a lower risk of falling compared to their single or unmarried counterparts. Being widowed was associated with an increased fall risk, highlighting the protective role of having a partner who provides social support [40]. These findings are consistent with other studies, including the South African census, which reported health advantages associated with marriage [41,42,43].

Medication use was significantly associated with falling. Similar to this study, previous research found that the use of anti-diabetic and anti-psychotic medications was associated with an increased fall risk [44,45]. Polypharmacy, or the use of multiple medications, was also identified as a significant risk factor for falls [14,22,46]. A systematic review further supported the finding that the use of four or more medications significantly increases the likelihood of falls. Reducing the use of psychotropic medications has been shown to decrease fall rates, highlighting the importance of careful medication management in fall prevention [47,48].

## 5. Conclusions

This study revealed a high incidence falls among older adults living in LTC facilities in the CoCT, with one-third of participants experiencing such events. These findings align with previous research, highlight the urgent need for interventions to reduce injuries and enhance the well-being of older adults. As participants age, the occurrence of falls increases, particularly among those engaging in high-risk behaviors. Despite technological advancements, a thorough history-taking process remains essential in assessing fall risk among older adults.

By identifying and understanding the factors that elevate the risk of falls, healthcare professionals can play a vital role in reducing the incidence of falls. This highlights the importance of examining the roles of long-term care staff in ensuring that the appropriate healthcare professionals are designated to specific tasks, ensuring that older adults in these facilities receive the necessary care to address this global concern [49]. Future research should focus on interventions aimed at promoting educational and behavioral changes among older adults to reduce the risk of falls and related injuries.

## Figures and Tables

**Figure 1 ijerph-22-00432-f001:**
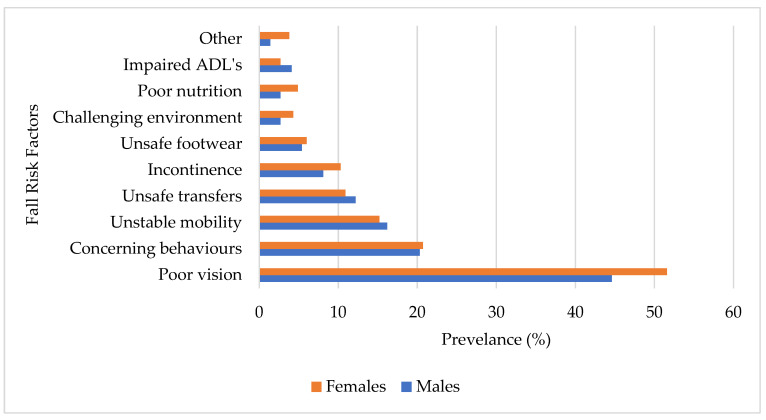
Fall-risk factors based on gender.

**Figure 2 ijerph-22-00432-f002:**
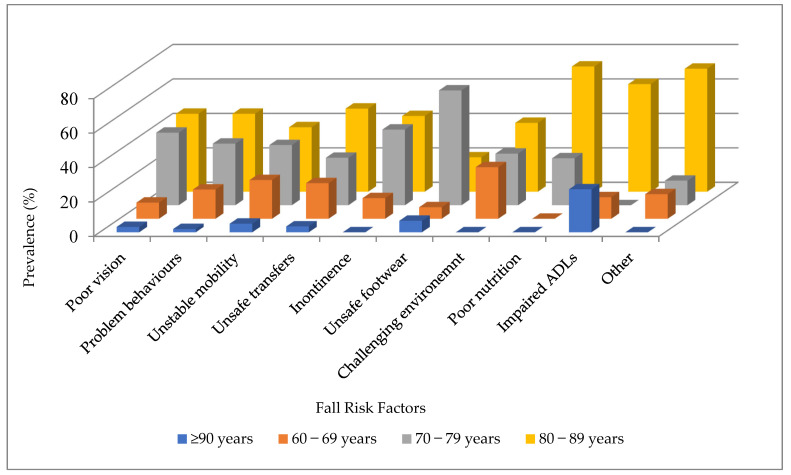
Fall-risk factors based on age group. Note: ADLs indicates activities of daily living.

**Figure 3 ijerph-22-00432-f003:**
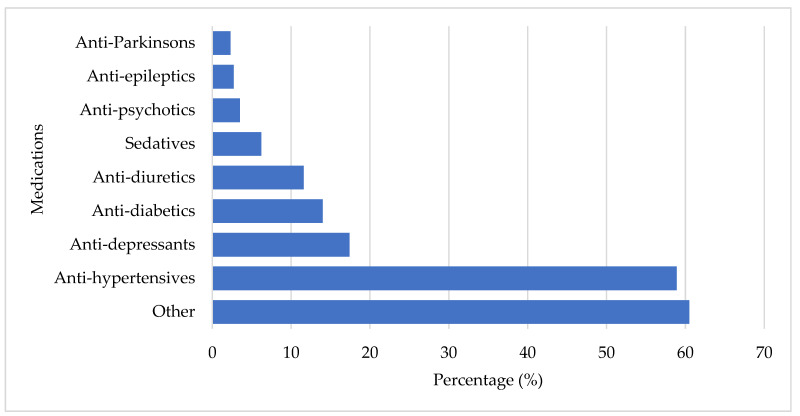
Medications used by the participants.

**Table 1 ijerph-22-00432-t001:** Sociodemographic characteristics of the participants.

Category	*n* (%)
**Gender**	
Male	74 (28.7)
Female	184 (71.3)
Total	258 (100)
**Age (years)**	
60–69	34 (13.2)
70–79	103 (39.9)
80–89	111 (43.0)
≥90	10 (3.9)
Total	258 (100)
**Obesity (kg/m^2^)**	
Underweight	10 (3.9)
Normal	113 (43.8)
Overweight	87 (33.7)
Obese	48 (18.6)
**Educational Level**	
Grade 7 or lower	32 (12.4)
Grades 8–11	43 (16.7)
Matriculated from mainstream school	79 (30.6)
Matriculated from technical school	51 (19.8)
Graduated with a bachelor’s degree	42 (16.3)
Graduated with a master’s degree	6 (2.3)
Graduated with a doctoral degree	5 (1.9)
Total	258 (100)
**Marital Status**	
Married	63 (24.4)
Single	44 (17.1)
Divorced	39 (15.1)
Widowed	112 (43.4)
Total	258 (100)
**Fall History**	
No	67.4 (174)
Yes	32.6 (84)
Total	258 (100)
**Fall Mechanisms**	
Slipping/tripping	40 (15.5)
Loss of balance	27 (10.5)
Collapse	9 (3.5)
Legs give way	2 (0.8)
Dizziness	13 (5.0)
Total	258 (100)

**Table 2 ijerph-22-00432-t002:** Relationship between the sociodemographic characteristics of participants and falls.

Variable	Category	Falls *n* (%)		*p*-Value	X^2^
		Yes	No		
Facility type	Non-profit organization	58 (39.4)	89 (55.1)	0.007 *	7.403
	Private	26 (23.4)	85 (80.1)		
Gender	Male	23 (31.0)	51 (68.9)	0.748	0.103
	Female	61 (33.1)	123 (66.8)		
Age (years)	60–69	12 (35.2)	22 (64.7)	0.806	0.982
	70–79	35 (33.9)	68 (66.0)		
	80–89	35 (31.5)	76 (68.4)		
	≥90	2 (20)	8 (80)		
Educational level	Junior school or lower	19 (59.3)	13 (40.6)	0.029 *	14.05
	Some high school, did not graduate	11 (25.5)	32 (74.4)		
	Matriculated from mainstream school	24 (30.3)	55 (69.6)		
	Technical school	12 (23.5)	39 (76.4)		
	Bachelor’s degree	15 (35.7)	27 (64.2)		
	Master’s degree	2 (33.3)	4 (66.6)		
	Doctoral degree	1 (20)	4 (80)		
Marital status	Married	16 (25.3)	47 (74.6)	0.001 *	16.49
	Single	20 (45.4)	24 (54.5)		
	Divorced	21 (53.8)	18 (46.1)		
	Widowed	27 (24.1)	85 (75.8)		

**Note:** * indicates statistically significant association *p* < 0.05.

## Data Availability

The fall-risk assessment tool data can be found at https://www.health.vic.gov.au/publications/falls-risk-assessment-tool-frat (accessed on 1 February 2022).

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
