# Peer review of "The Prevalence of Falls Among Older Adults Living in Long-Term Care Facilities in the City of Cape Town"

_ijerph, 2025, doi:10.3390/ijerph22030432_

Round 1

Reviewer 1 Report

Comments and Suggestions for Authors

See attached file.

Author Response

Dear reviewer, please see attached file.

Reviewer 2 Report

Comments and Suggestions for Authors

The subject matter of falls among the elderly is important, and there is under-reporting from developing countries such as South Africa.  That credit granted, the manuscript is significantly underdeveloped as to be underwhelming.  First, the tile “ The Prevalence of Falls Among Older Adults Living in Long 2 Term Care Facilities” does not match the contents of the manuscript.  As stated, the reader expected a report of findings on types of falls among the elderly, which is not even reported on this manuscript. Turns out that the manuscript is on bio-behavioral correlates  of falls among the elderly (going by the data collected reporting of findings).  Even granted that modification, the authors fail to present an introduction on bio-behavioural correlates to make a case on which of these would be more prevalent in that population and setting, including why and how. The authors would then make an evidence informed decision on which factors may be under-researched  their setting.  The authors completely miss the boat,  and say to be investigating
“associations between facility type, marital status, educational level, 19 and their relationship to falling”, of which they do not present any literature review on those factors (other than they are not what the study is about). Later in the manuscript they say to report on bio-behavioral correlates of falls “such as agitation or confusion, and other unmentioned risk factors. antidepressant and antidiabetic medications, medications used, and the participant’s psychological and cognitive status,  and intrinsic factors (e.g., visual problems, osteoporosis, balance difficulties), extrinsic 51 (e.g., environmental hazards), and behavioural factors.  This should be the focus of the study with re-titling and a complete re-do of the introduction section. Overall a high prospect study but badly conceived.

Author Response

(The authors gave the same response as above.)

Reviewer 3 Report

Comments and Suggestions for Authors

Line 32-35: why institutionalized elderly is more likely to fall, please state reasons

Introduction is well framed
line 66: write the hypothesis as well after the aim

Line 68: from which area or location they were recruited. Please mention.

Line 72-75. These sentences are better suited at the beginning

Line 75: Why do you choose only 60-year-olds and above? Why not 65 years’ old, which would be considered elderly, and why not from 50, when most of the women suffer from menopause, ostephoeresis, and falls?

Line 82: provide reliability and validity of FRAT

Line 94: Add procedure as a separate heading like who gave them the questionnaire, how the data was collected, how long the participants took to answer, who collected the data, and so on. Make sure the procedure is replicable.

Line 100: Did you conduct assumption testing for the data? Please mention the assumption testing was conducted or not.

Line 103: What do you mean that initially 357 participants consented and only 258 participated? They were supposed to fill out a questionnaire personally. Why did COVID-19 stop you?

 Statistical analysis: Please add the FRAT score as dependent variable and conduct the linear regression to determine the influence of each demographic variable on the risk of falls

Line 145: Even if the figure gives us a rough idea about the fall, regression analysis will be more robust.

Line 163: The statistically significant difference is between fallers or non-fallers? There has to be two p values. What is compared with what?

Author Response

Dear reviewer, please see attached filed. 

Round 2

Reviewer 1 Report

Comments and Suggestions for Authors

I am pleasantly surprised and grateful for the quality with which all the errors have been corrected, good job. There are only a couple of things left to improve, and they are the formatting of the tables, as the instructions for authors indicate, the tables should only have the top and bottom lines of the complete table and the bottom of the first row. There should only be more horizontal lines than indicated if they improve the comprehension of the table, so correct the tables in this way.
Also, I noticed that there are only two tables and the second one is called table 3, correct this title and make the correction every time you mention this table in the text.

Reviewer 2 Report

Comments and Suggestions for Authors

This is good to publish with in-house editing as may be needed

Reviewer 3 Report

Comments and Suggestions for Authors

All my comments are answered satisfactorily by the authors
